# Cytochalasin B Mitigates the Inflammatory Response in Lipopolysaccharide-Induced Mastitis by Suppressing Both the ARPC3/ARPC4-Dependent Cytoskeletal Changes and the Association Between HSP70 and the NLRP3 Inflammasome

**DOI:** 10.3390/ijms26073029

**Published:** 2025-03-26

**Authors:** An Wang, Yan Chen, Bo Fang, Jiang Zhang, Wenkai Bai, Tingji Yang, Quanwei Zhang, Peiwen Liu, Zhiwei Duan, Ting Lu, Yuxuan He, Yong Zhang, Xingxu Zhao, Weitao Dong

**Affiliations:** 1College of Veterinary Medicine, Gansu Agricultural University, Lanzhou 730070, China; 1073323120304@st.gsau.edu.cn (A.W.); xyzchenyan@163.com (Y.C.); fangb_ioi@163.com (B.F.); zj_m80@163.com (J.Z.); bwk20001129@163.com (W.B.); ytj19980227@163.com (T.Y.); 18893134810@163.com (P.L.); 15002514846@163.com (Z.D.); asdlting@163.com (T.L.); heyuxuan1226@126.com (Y.H.); zhang1234y56@163.com (Y.Z.); zhaoxx@gsau.edu.cn (X.Z.); 2Gansu Key Laboratory of Animal Generational Physiology and Reproductive Regulation, Lanzhou 730070, China; zhangqw@gsau.edu.cn; 3College of Life Sciences and Biotechnology, Gansu Agricultural University, Lanzhou 730030, China

**Keywords:** dairy cow, mastitis, cytochalasin B, cytoskeletal rearrangement, NLRP3 inflammasome

## Abstract

Cow mastitis is a major challenge in dairy farming, significantly affecting both milk quality and cow health. Cytochalasin B (CB) is a fungal toxin and an actin cytoskeleton depolymerizing agent that exhibits anti-inflammatory and antitumor properties; however, its mechanism in cow mastitis remains unclear. In this study, we systematically evaluated the effects of CB on mastitis using an LPS-induced inflammation model in bovine mammary epithelial cells (MAC-T) and a mouse mastitis model. The techniques employed included Real-time quantitative PCR detecting system (qPCR), Western blot, HE staining, immunofluorescence (IF), and immunohistochemistry (IHC). The results demonstrated that CB significantly alleviated LPS-induced mastitis by downregulating the expression of pro-inflammatory factors IL-1β, TNF-α, and the NLRP3 inflammasome while also reducing cell apoptosis. Further mechanistic investigations revealed that CB mitigates the inflammatory response by inhibiting the expression of ARPC3, ARPC4, and HSP70, thereby disrupting cytoskeletal rearrangement and the activation of the NLRP3 inflammasome. Overall, this study reveals the potential therapeutic role of CB in cow mastitis and provides a theoretical foundation for developing novel intervention strategies.

## 1. Introduction

Bovine mastitis is one of the most common diseases in the global dairy industry, posing a serious threat to cow health and the quality of dairy products. According to estimates by the Food and Agriculture Organization of the United Nations (FAO, 2020), mastitis causes annual economic losses amounting to billions of dollars worldwide due to reduced milk yield, increased treatment costs, and expenses associated with culling cows [1]. Pathogenic microbial infections are the primary cause of mastitis, with lipopolysaccharide (LPS) from Gram-negative bacteria playing a critical role in triggering the disease by eliciting a robust inflammatory response [2,3]. Although antibiotics are currently the main treatment for bovine mastitis, the increasing prevalence of antibiotic-resistant strains and concerns over antibiotic residues have prompted researchers to explore novel therapeutic strategies.

Cytochalasin B (CB) is a mycotoxin derived from Fusarium species, with a molecular weight of approximately 479.6 Da. As an actin depolymerizer, CB specifically binds to the growing ends of actin filaments, preventing the polymerization of actin monomers and disrupting the dynamic equilibrium of the cytoskeleton [4]. Recent studies have demonstrated that CB possesses significant anti-inflammatory and antitumor properties. Regarding its anti-inflammatory effects, CB attenuates inflammation by inhibiting the migration of inflammatory cells and reducing the release of inflammatory mediators; for example, it plays a crucial role in modulating macrophage-mediated inflammatory responses [5]. Concerning its antitumor activity, CB disrupts the cytoskeletal framework of tumor cells, thereby suppressing their proliferation and invasiveness; for instance, in gliomas, CB induces apoptosis in tumor cells [6], and in hepatocellular carcinoma (HCC), it shows promise as a potential therapeutic agent against this highly invasive tumor [7].

The actin cytoskeleton plays a pivotal role in regulating immune function by orchestrating fundamental cellular processes such as proliferation, differentiation, apoptosis, migration, and signal transduction [8]. Research has shown that remodeling of the actin cytoskeleton is essential not only for the function of immune cells—such as T lymphocytes, B lymphocytes, and macrophages—but also for the broader regulation of the immune system [9]. In the context of inflammation, the pro-inflammatory factor LPS induces dynamic cytoskeletal changes through activation of the PI3K signaling pathway [10,11]. This process relies on actin polymerization mediated by the actin-related protein 2/3 (ARP2/3) complex, which directly affects the migratory and phagocytic capabilities of inflammatory cells, thereby intensifying the inflammatory response [12].

The ARP2/3 complex plays a key role in LPS-induced inflammatory responses. Composed of ARP2, ARP3, and five subunits (ARPC1, ARPC2, ARPC3, ARPC4, and ARPC5), this complex is stably assembled, with ARPC4 being crucial for maintaining its structural integrity and stability through specific interactions with ARPC3 [13,14]. During inflammation, the ARP2/3 complex, activated by neuronal Wiskott–Aldrich syndrome protein (N-WASP), promotes the formation of branched actin structures, thereby enhancing the migratory and phagocytic capabilities of inflammatory cells [12]. Moreover, studies have observed that during bacterial invasion, the expression levels of ARPC3 and ARPC4 increase significantly, and inhibiting their expression can effectively mitigate the host’s inflammatory response [15].

Furthermore, additional critical factors are involved in the molecular mechanisms underlying inflammation. For example, fever—a hallmark of inflammation—is primarily driven by interleukin-1β (IL-1β), whose production depends on the activation of the NLRP3 inflammasome [16]. However, excessive activation of the NLRP3 inflammasome can lead to tissue damage and immune dysfunction, contributing to the onset and progression of inflammatory diseases such as mastitis [17,18]. Heat shock protein HSP70, which functions as a molecular chaperone, interacts with the NLRP3 inflammasome and acts as a negative regulator of its activation. Additionally, HSP70 helps maintain cellular function by participating in clathrin (CLTC)-mediated endocytosis via N-WASP [19,20]. Notably, HSP70 plays dual roles in inflammation: while its elevated expression can suppress apoptosis and enhance host immune responses [21], HSP70 induced by the heat shock response following pro-inflammatory stimuli may exert cytotoxic effects, exacerbating tissue damage [22].

Previous studies by our team have identified a potential interaction mechanism between ARPC3/4 and HSP70 in an LTA-induced bovine mastitis inflammation model [23]. Building on these findings, the current study employs cytochalasin B (CB) as a potential anti-inflammatory agent, using LPS to establish inflammatory models in MAC-T cells and mice. This research investigates the anti-inflammatory mechanisms of CB from two perspectives: first, CB may modulate cytoskeletal rearrangement by inhibiting the expression of ARPC3, ARPC4, and HSP70, thereby alleviating LPS-induced inflammation in bovine mammary epithelial cells; second, CB may disrupt the interaction between HSP70 and the NLRP3 inflammasome, blocking its activation and subsequently attenuating the LPS-induced inflammatory response. Overall, this study aims to provide novel mechanistic insights and potential intervention strategies for the treatment of bovine mastitis.

## 2. Results

### 2.1. Expression of Inflammatory Cytokines and Activation of the NLRP3 Inflammasome in Bovine Mastitis Tissue

To assess the immune environment in bovine mammary tissue, qPCR and Western blot were used to evaluate the expression of IL-1β and TNF-α, as well as the activation of the NLRP3 inflammasome, in tissues from healthy cows (control) and those with clinical (CM) and subclinical mastitis (SCM). Compared with the control group, both the CM and SCM groups showed significantly elevated levels of IL-1β, TNF-α, and NLRP3 (Figure 1).

### 2.2. Increased Apoptosis in Bovine Mastitis Tissue

Given that inflammation is typically accompanied by enhanced apoptosis, qPCR and Western blot were used to measure the expression of Bax, Bcl-2, Caspase 3, and Caspase 7 in mammary tissues from the control, CM, and SCM groups. The results revealed that, relative to the control group, Bcl-2 expression was significantly decreased, while the Bax, Caspase 3, and Caspase 7 levels were markedly increased in both the CM and SCM groups (Figure 2).

### 2.3. Enhanced Expression of ARPC3, ARPC4, and HSP70 in Bovine Mastitis Tissue

To examine changes in ARPC3, ARPC4, and HSP70 expression in bovine mastitis, qPCR and Western blot were used to determine the levels of ARPC3, ARPC4, HSP70A1A, and HSP70A1L in tissues from healthy cows (control) and cows with CM and SCM. Both mRNA and protein levels of these markers were significantly elevated in the CM and SCM groups compared with the control group (Figure 3).

### 2.4. CB Inhibits LPS-Induced Expression of Inflammatory Cytokines and Activation of the NLRP3 Inflammasome in MAC-T Cells

CB has been shown to have potent anti-inflammatory effects. To investigate its impact on LPS-induced inflammatory responses in MAC-T cells, qPCR and Western blot were used to assess the mRNA and protein expression of IL-1β, TNF-α, and the NLRP3 inflammasome. The LPS + CB group exhibited significantly lower levels of these inflammatory markers compared with the LPS group (Figure 4).

### 2.5. CB Inhibits LPS-Induced Apoptosis in MAC-T Cells

To explore the role of CB in regulating apoptosis in LPS-stimulated MAC-T cells, qPCR, Western blot, and immunofluorescence (IF) were employed to detect apoptosis-related proteins. The LPS + CB group showed a significant increase in Bcl-2 expression and a significant decrease in Bax, Caspase 3, and Caspase 7 expression compared with the LPS group (Figure 5).

### 2.6. CB Inhibits LPS-Induced Expression of ARPC3, ARPC4, and HSP70 in MAC-T Cells

To further elucidate the anti-inflammatory mechanism of CB, qPCR, Western blot, and IF were used to examine the expression of ARPC3, ARPC4, HSP70A1A, and HSP70A1L in MAC-T cells. The results demonstrated that the LPS + CB group had a significantly lower expression of these markers compared with the LPS group, indicating that CB effectively suppresses their LPS-induced expression (Figure 6).

### 2.7. CB’s Effects on LPS-Induced Pathological Damage and Inflammatory Responses in Mouse Mastitis Tissue

To further investigate the anti-inflammatory effects of CB, mouse mammary tissues were subjected to HE staining, immunohistochemistry (IHC), and IF to observe IL-1β, TNF-α, and NLRP3. LPS treatment disrupted tissue structure, as evidenced by increased stroma thickness, rupture, and even atrophy and necrosis of glandular alveoli, along with the detachment of mammary epithelial cells and extensive inflammatory cell infiltration. However, CB treatment markedly ameliorated these pathological changes. Additionally, qPCR and Western blot confirmed that LPS significantly increased IL-1β, TNF-α, and NLRP3 expression in mouse mammary tissues compared with the control group, while CB treatment mitigated these effects (Figure 7). These findings confirm the successful establishment of the mouse mastitis model and demonstrate that CB alleviates mastitis by inhibiting inflammatory cytokine production.

### 2.8. CB Inhibits LPS-Induced Apoptosis in Mouse Mammary Cells

To further explore CB’s regulatory effects on apoptosis in inflammatory conditions, qPCR, Western blot, IHC, and IF were used to assess apoptosis-related protein expression in mouse mammary cells. The LPS + CB group showed a significant increase in Bcl-2 expression and significant decreases in Bax, Caspase 3, and Caspase 7 expression compared with the LPS group (Figure 8).

### 2.9. CB Inhibits LPS-Induced Expression of ARPC3, ARPC4, and HSP70 in Mouse Mammary Tissue

To further elucidate the anti-inflammatory mechanism of CB, qPCR, Western blot, IHC, and IF were used to measure the expression of ARPC3, ARPC4, HSP70A1A, and HSP70A1L in mouse mammary tissues. The LPS + CB group exhibited significantly lower levels of these proteins compared with the LPS group, indicating that CB effectively inhibits their LPS-induced expression (Figure 9).

## 3. Discussion

Bovine mastitis is a major challenge in the dairy industry, severely affecting milk quality and cow health while incurring significant economic losses. Traditionally treated with antibiotics, mastitis management faces issues such as adverse side effects and growing antibiotic resistance, prompting the search for alternative therapies [24,25]. This study systematically investigates the mechanism of cytochalasin B (CB) in LPS-induced mastitis through both in vitro and in vivo experiments. Our results demonstrate that CB significantly alleviates mastitis symptoms by suppressing inflammatory responses, inhibiting apoptosis, and downregulating key molecules including ARPC3, ARPC4, and HSP70, thereby providing a theoretical basis for its potential therapeutic application.

Initially, we assessed the expression of inflammatory cytokines IL-1β, TNF-α, and the activation of the NLRP3 inflammasome in bovine mammary tissues using qPCR and Western blot. Both mRNA and protein levels of these factors were significantly elevated in tissues from cows with clinical (CM) and subclinical mastitis (SCM) compared with healthy controls. This observation aligns with previous research demonstrating that LPS, a critical pathogenic component of Gram-negative bacteria, activates immune regulatory pathways and the NLRP3 inflammasome, thereby promoting IL-1β release and exacerbating inflammatory damage [26,27,28]. Notably, CB treatment in MAC-T cells and mouse mastitis models markedly reduced the expression of IL-1β, TNF-α, and NLRP3, indicating potent anti-inflammatory effects similar to those observed with other natural compounds such as allicin and lentinan [29,30].

Inflammation is frequently accompanied by increased apoptosis, a phenomenon particularly evident in mastitis pathology [31,32]. Our study found that pro-apoptotic proteins Bax, Caspase 3, and Caspase 7 were significantly upregulated, while the anti-apoptotic protein Bcl-2 was downregulated in the CM and SCM groups, indicating heightened apoptosis. In both LPS-induced MAC-T cells and mouse models, CB treatment reversed these changes by upregulating Bcl-2 and downregulating Bax, Caspase 3, and Caspase 7. As an actin depolymerizer, CB may protect cells by disrupting cytoskeletal structures and blocking apoptotic signaling pathways, a mechanism consistent with reports in other inflammatory conditions [33].

To further elucidate CB’s anti-inflammatory mechanism, we examined the expression of ARPC3, ARPC4, and HSP70. Our data revealed that the ARPC3, ARPC4, and HSP70 levels were significantly elevated in the CM and SCM groups but markedly reduced in CB-treated MAC-T cells and mouse models. ARPC3 and ARPC4 are critical components of the ARP2/3 complex, which regulates actin polymerization and cytoskeletal rearrangement, processes that facilitate the migration and phagocytic activity of inflammatory cells [12,34,35]. By downregulating ARPC3 and ARPC4, CB likely disrupts cytoskeletal rearrangement, thereby reducing inflammatory cell migration and attenuating inflammation.

HSP70, a molecular chaperone, plays a complex role in inflammation regulation by interacting with the NLRP3 inflammasome. Although generally considered a negative regulator of NLRP3 activation [19], HSP70 overexpression following pro-inflammatory stimuli can have cytotoxic effects, contributing to apoptosis in endotoxin-exposed cells [36,37]. In our study, CB reduced HSP70 expression, which may decrease its interaction with NLRP3 and further inhibit inflammasome activation. This finding supports the notion that CB exerts its anti-inflammatory effects, at least in part, through the HSP70-NLRP3 axis.

In summary, CB significantly mitigates inflammatory damage in LPS-induced mastitis models by suppressing inflammatory cytokine production, inhibiting apoptosis, and downregulating ARPC3, ARPC4, and HSP70 expression. Its mechanism involves disrupting cytoskeletal rearrangement and blocking NLRP3 inflammasome activation. As a natural product, CB may offer advantages over traditional antibiotics, including lower toxicity and reduced residue risks, aligning with current trends in mastitis treatment research. However, further studies are needed to clarify CB’s specific molecular targets and dose–response relationships, and to optimize its application conditions for translational use in bovine mastitis therapy.

## 4. Materials and Methods

### 4.1. Cell Culture and Treatment

Bovine mammary epithelial (MAC-T) cells were obtained from the Chinese Academy of Agricultural Sciences. The cells were cultured in DMEM/F12 medium supplemented (Gibco, Grand Island, NY, USA) with 10% fetal bovine serum (Invigentech, Irvine, CA, USA) in a 37 °C and 5% CO_2_ incubator. They were maintained in 25 cm^2^ culture flasks, and experiments were initiated when cell confluence reached 60–70%. For treatment, LPS (Sigma, St. Louis, MO, USA) was dissolved in sterile PBS (Servicebio, Wuhan, China) and applied at a final concentration of 50 µg/mL for 24 h. Cytochalasin B (CB) was purchased from MedChemExpress (MCE, Monmouth Junction, NJ, USA). CB was dissolved in 1 mL of dimethyl sulfoxide (DMSO) (Servicebio, Wuhan, China) to prepare a 1 mg/mL stock solution and used at a final concentration of 500 ng/mL. After the initial 24 h LPS treatment, CB was added to the medium, and the cells were further incubated for an additional 24 h.

### 4.2. Tissue Sample Preparation

All cows were sourced from a large, standardized dairy farm in Wuzhong, Ningxia, and were all at the same lactation stage. A resident veterinarian conducted diagnoses to exclude other diseases. Udder health was evaluated based on criteria such as redness, swelling, heat, and hardness, along with somatic cell count (SCC) and the Lanzhou Mastitis Test (LMT). Healthy cows exhibited normal udder skin and milk, with SCC values between 7 × 10^4^ and 1 × 10^5^ cells/mL and negative LMT results. Cows with subclinical mastitis, despite lacking overt clinical symptoms, showed abnormal milk test results with SCC values ranging from 2 × 10^5^ to 5 × 10^5^ cells/mL and weakly positive LMT results (+ or ++). In contrast, cows with clinical mastitis displayed clear signs such as udder redness, swelling, and heat, with milk showing watery consistency, clots, or blood; these cows had SCC values between 13 × 10^5^ and 15 × 10^5^ cells/mL and positive LMT results (+++). We selected three healthy Holstein cows (control group, Con, n = 3), three cows with subclinical mastitis (subclinical group, SM, n = 3), and three cows with clinical mastitis (clinical group, CM, n = 3) and sent them to the slaughterhouse. After slaughter, fresh udder tissue from each group was collected under sterile conditions for research. Portions of the tissue were cut into 1.5 cm^3^ blocks and fixed in sterile, enzyme-free centrifuge tubes containing 4% paraformaldehyde, while the remaining tissue was cut into 1 cm^3^ blocks and stored in sterile, enzyme-free cryovials at −80 °C for future use.

### 4.3. Animal Housing and Treatment

Fifty 8-week-old female Kunming mice and twenty-five male Kunming mice were obtained from the Chinese Academy of Agricultural Sciences. After a 7-day acclimation period with free access to food and water, the mice were housed at a 2:1 female-to-male ratio for a 2-day mating period. Eighteen pregnant mice with similar physical conditions were then selected, and following 7 days of lactation, they were randomly assigned to three groups: control, LPS, and CB + LPS. The fourth pair of mammary glands was injected as follows: the LPS and CB + LPS groups received a 50 µL injection of LPS solution (200 µg/mL) to induce mastitis, while the control group was injected with an equal volume of saline. Twenty-four hours after the LPS injection, the CB + LPS group was administered CB (4 mg/kg). After an additional 24 h, the fourth pair of mammary gland tissues was collected under sterile conditions. Tissue samples were divided, with one portion cut into 1.5 cm^3^ blocks and stored at −80 °C, and the remainder were fixed in 4% formaldehyde for subsequent analyses. This study was approved by the Animal Protection Committee of Gansu Agricultural University.

### 4.4. RNA Extraction, cDNA Synthesis, and qPCR

Total RNA was isolated from cells and tissues using TRNzol Universal Reagent (G3013, Servicebio, Wuhan, China) according to the manufacturer’s protocol. RNA concentration and purity were measured with a NanoDrop ND-1000 spectrophotometer (Thermo Fisher, San Diego, CA, USA); only samples with an OD260/280 ratio between 1.8 and 2.0 were used. cDNA was synthesized from total RNA using the Evo M-MLV Reverse Transcription Kit (AG11732, Accurate Biotechnology, Changsha, China). Gene expression was quantified with SYBR Green Pro Taq HS Premix (AG11746, Accurate Biotechnology, Changsha, China), using β-actin as the internal reference gene. Relative expression levels were calculated using the 2^−ΔΔCT^ method. mRNA-specific primer sequences are listed in Table 1.

### 4.5. Western Blot

Total protein was extracted from cells and animal mammary tissues using RIPA lysis buffer (Solarbio, Beijing, China) supplemented with PMSF at a 100:1 ratio. Protein concentration was measured with a BCA assay (PC0020, Solarbio, Beijing, China). The supernatant was mixed with 5× loading buffer, heat-denatured at 98 °C for 15 min, and stored at −80 °C. Western blotting was performed to assess the expression of IL-1β, TNF-α, NLRP3, Bax, Bcl-2, Caspase 3, Caspase 7, ARPC3, ARPC4, HSP70A1A, and HSP70A1L. Equal amounts of protein (50 μg) were separated by SDS-PAGE and transferred to a PVDF membrane. The membrane was washed with 1× PBST, blocked in 5% non-fat milk at room temperature for 2 h, and then incubated overnight at 4 °C with the appropriate primary antibody. The following day, the membrane was washed three times with PBST (10 min each), incubated with the corresponding secondary antibody at 37 °C for 1 h, and washed again three times with PBST. Protein bands were visualized using a chemiluminescent solution with β-actin as the internal control, and band intensities were quantified using Image-J 1.52a software. The antibodies used are listed in Table 2.

### 4.6. Immunohistochemistry and Immunofluorescence Staining

Mouse mammary tissues preserved in formaldehyde were dehydrated, paraffin-embedded, sectioned into 4 µm slices, and deparaffinized using graded alcohols and xylene before hematoxylin–eosin (HE) staining. After dehydration and coverslipping, the slides were examined under a microscope (Axiocam 208 color, Zeiss, Oberkochen, Germany). For immunohistochemistry (IHC), antigen retrieval was performed with heated citrate buffer and sections were blocked with 5% BSA. The primary antibody was applied overnight at 4 °C, followed by a 1 h incubation with the secondary antibody at 37 °C, then DAB staining and hematoxylin counterstaining were performed, after which the slides were coverslipped for observation. For immunofluorescence (IF), MAC-T cells and mouse mammary tissue sections were stained using a three-marker, four-color multiplex fluorescent kit (AiFang Biological, Changsha, China) per the manufacturer’s instructions and observed with an inverted fluorescence microscope (Revolve Omega, ApexBio, Suzhou, China).

### 4.7. Statistical Analysis

Western blot bands were quantified using ImageJ software. Statistical analyses were performed with SPSS 25.0, and all data are presented as mean ± standard deviation. A *p* value < 0.05 (*) was considered statistically significant, while *p* < 0.01 (**) was deemed highly significant. GraphPad Prism 8.0.2 was used for data visualization.

## 5. Conclusions

In both the LPS-induced MAC-T cell inflammation model and the mouse mastitis model, CB effectively inhibited inflammatory damage in vitro and in vivo. Furthermore, CB exerts its anti-inflammatory effects by suppressing the expression of ARPC3, ARPC4, and HSP70, thereby disrupting cytoskeletal rearrangement and blocking NLRP3 inflammasome activation. These findings provide a novel theoretical foundation for the potential use of CB in treating bovine mastitis.

## Figures and Tables

**Figure 1 ijms-26-03029-f001:**
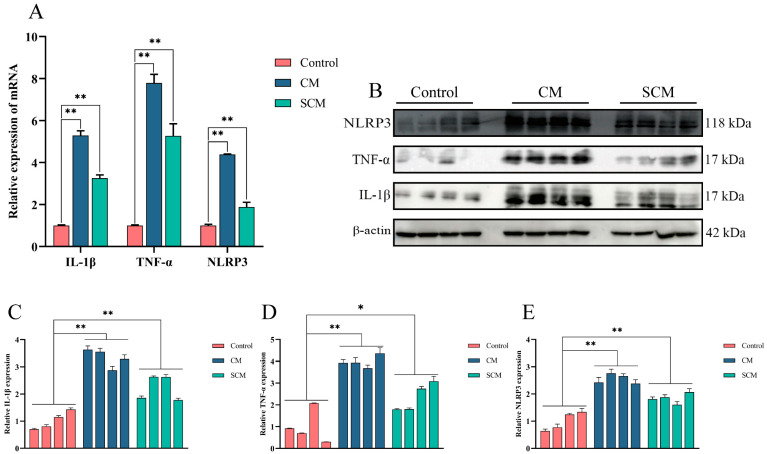
The expression of inflammatory cytokines and the activation of inflammasome in dairy cow mastitis tissue. (**A**) qPCR was used to detect the expression of inflammatory cytokines and NLRP3 inflammasome mRNA levels. (**B**–**E**) Western blot was used to detect the expression of inflammatory cytokines and NLRP3 inflammasome protein levels. *, *p* < 0.05; **, *p* < 0.01.

**Figure 2 ijms-26-03029-f002:**
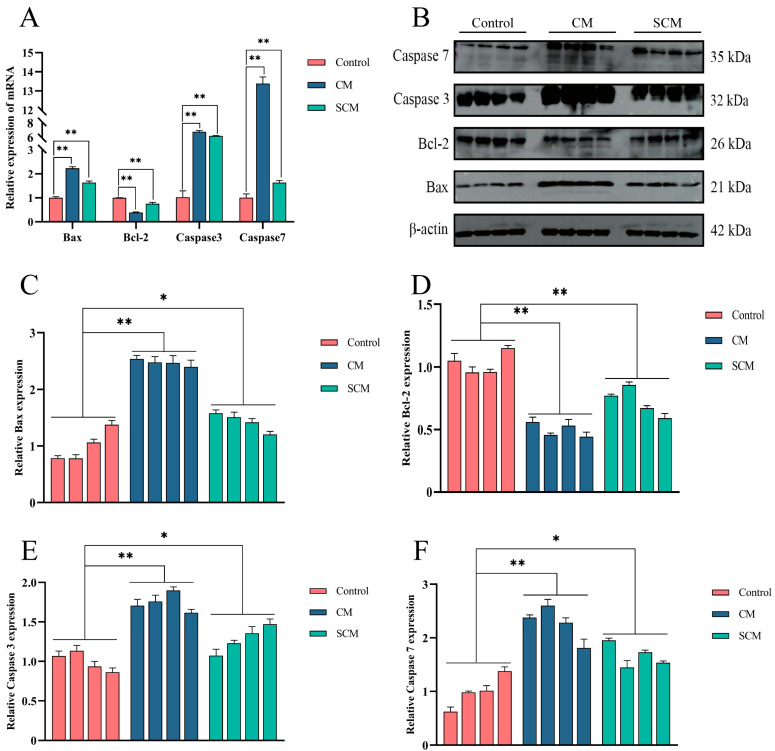
The apoptosis of inflammatory cells in dairy cow mammary gland was increased. (**A**) The mRNA levels of Caspase 3, Caspase 7, Bax, and Bcl-2 were detected by qPCR. (**B**–**F**) Western blot was used to detect the expression of Caspase 3, Caspase 7, Bax, and Bcl-2. *, *p* < 0.05; **, *p* < 0.01.

**Figure 3 ijms-26-03029-f003:**
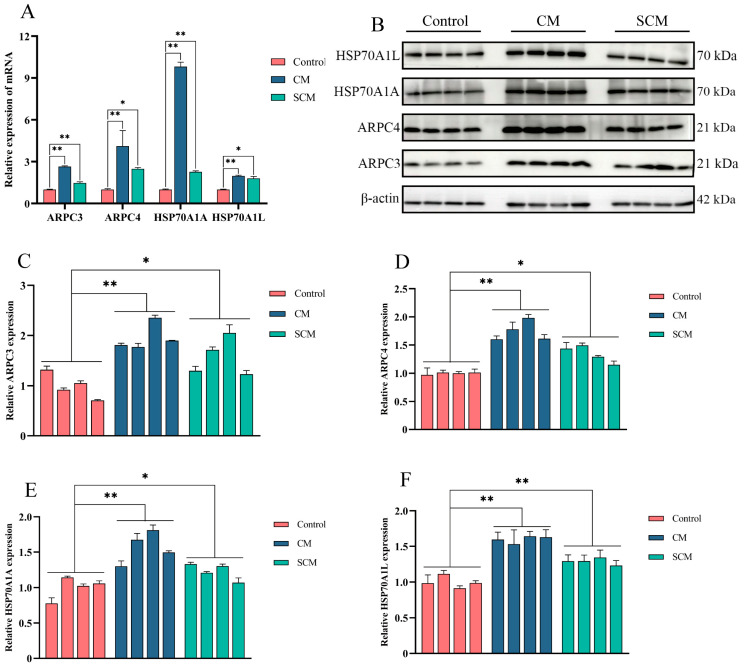
The expression of ARPC3, ARPC4, and HSP70 in mastitis tissues of dairy cows was enhanced. (**A**) The mRNA levels of ARPC3, ARPC4, HSP70A1A, and HSP70A1L were detected by qPCR. (**B**–**F**) Western blot was used to detect the expression of ARPC3, ARPC4, HSP70A1A, and HSP70A1L. *, *p* < 0.05; **, *p* < 0.01.

**Figure 4 ijms-26-03029-f004:**
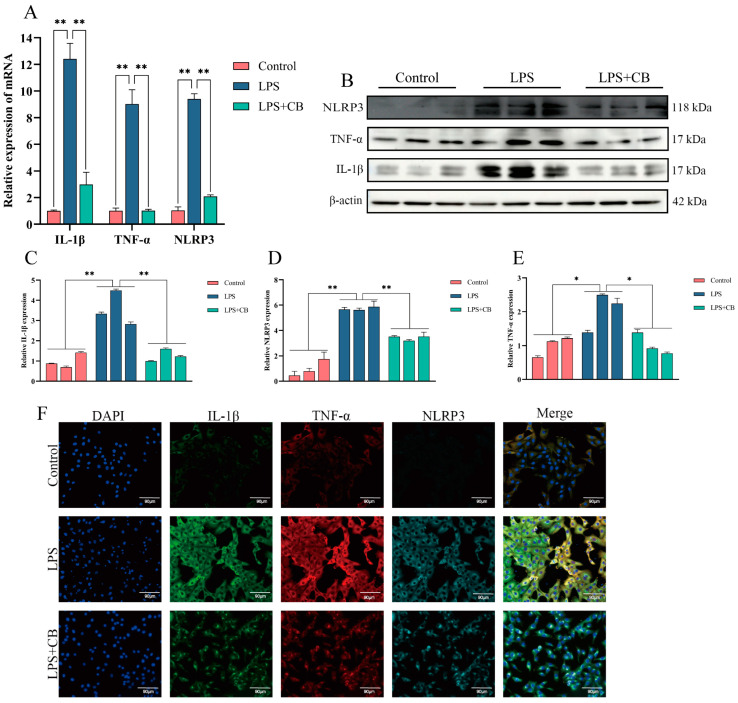
CB inhibited LPS-induced expression of inflammatory cytokines and activation of NLRP3 inflammasome in MAC-T cells. (**A**) qPCR was used to detect the expression of inflammatory cytokines IL-1β, TNF-α, and NLRP3 inflammasome mRNA levels. (**B**–**E**) Western blot was used to detect the expression of inflammatory cytokines IL-1β, TNF-α, and NLRP3 inflammasome protein levels. (**F**) IF was used to detect the expression of inflammatory cytokines IL-1β, TNF-α, and NLRP3 inflammasome protein. Scale bar: 90 μm (white line segment). *, *p* < 0.05; **, *p* < 0.01.

**Figure 5 ijms-26-03029-f005:**
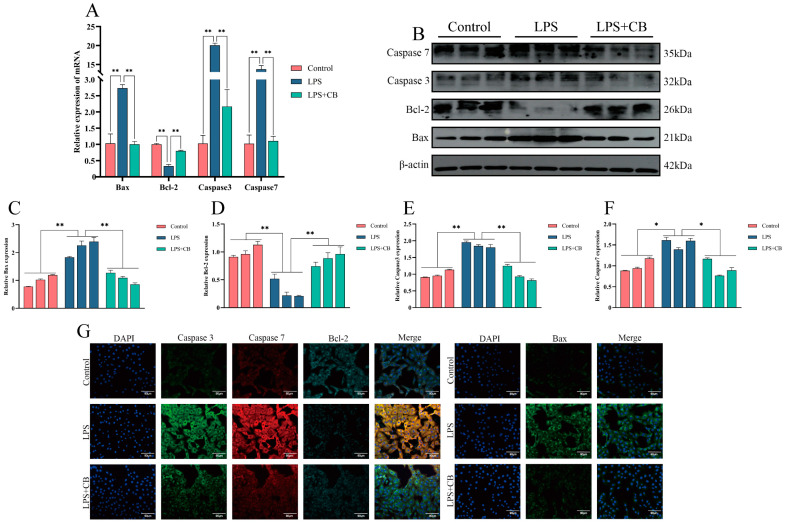
CB inhibited LPS-induced apoptosis of MAC-T cells. (**A**) The mRNA levels of Caspase 3, Caspase 7, Bax, and Bcl-2 were detected by qPCR. (**B**–**F**) Western blot was used to detect the expression of Caspase 3, Caspase 7, Bax, and Bcl-2. (**G**) The expression of Caspase 3, Caspase 7, Bax, and Bcl-2 protein was detected by IF. Scale bar: 90 μm (white line segment). *, *p* < 0.05; **, *p* < 0.01.

**Figure 6 ijms-26-03029-f006:**
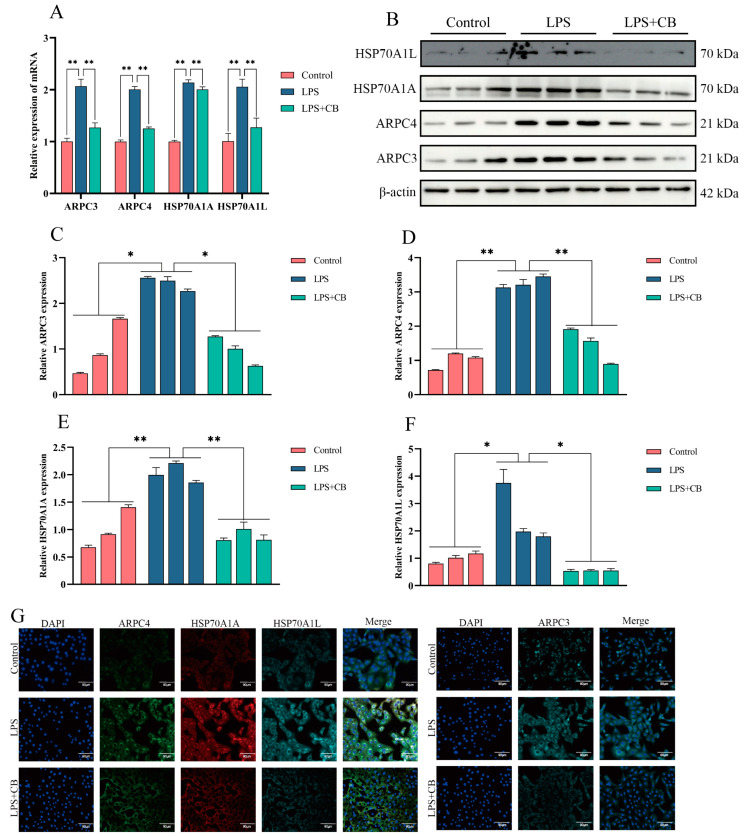
CB inhibited LPS-induced expression of ARPC3, ARPC4, and HSP70 in MAC-T cells. (**A**) The expression of ARPC3, ARPC4, HSP70A1A, and HSP70A1L mRNA levels was detected by qPCR. (**B**–**F**) Western blot was used to detect the expression of ARPC3, ARPC4, HSP70A1A, and HSP70A1L. (**G**) The expression of ARPC3, ARPC4, HSP70A1A, and HSP70A1L protein was detected by IF. Scale bar: 90 μm (white line segment). *, *p* < 0.05; **, *p* < 0.01.

**Figure 7 ijms-26-03029-f007:**
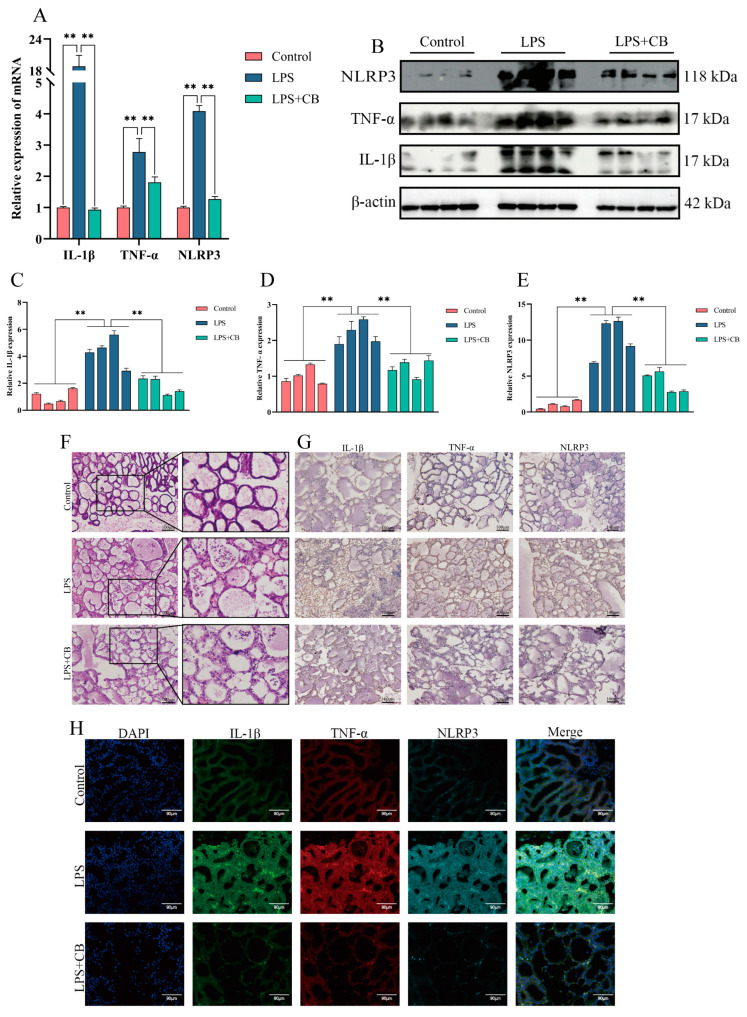
The effect of CB on the pathological damage of mammary gland in mice with LPS-induced mastitis and the effect of CB on the expression of inflammatory cytokines and the activation of NLRP3 inflammasome in mice with LPS-induced mastitis. (**A**) qPCR was used to detect the expression of inflammatory cytokines IL-1β, TNF-α, and NLRP3 inflammasome mRNA levels. (**B**–**E**) Western blot was used to detect the expression of inflammatory cytokines IL-1β, TNF-α, and NLRP3 inflammasome protein levels. (**F**) HE was used to observe the effect of CB on the pathological damage of mammary gland tissue in mice with LPS-induced mastitis. Scale bar: 100 μm (black line segment). (**G**,**H**) IHC and IF were used to detect the expression of inflammatory cytokines IL-1β, TNF-α, and NLRP3 inflammasome protein. Scale bar: 90 μm (white line segment),100 μm (black line segment). **, *p* < 0.01.

**Figure 8 ijms-26-03029-f008:**
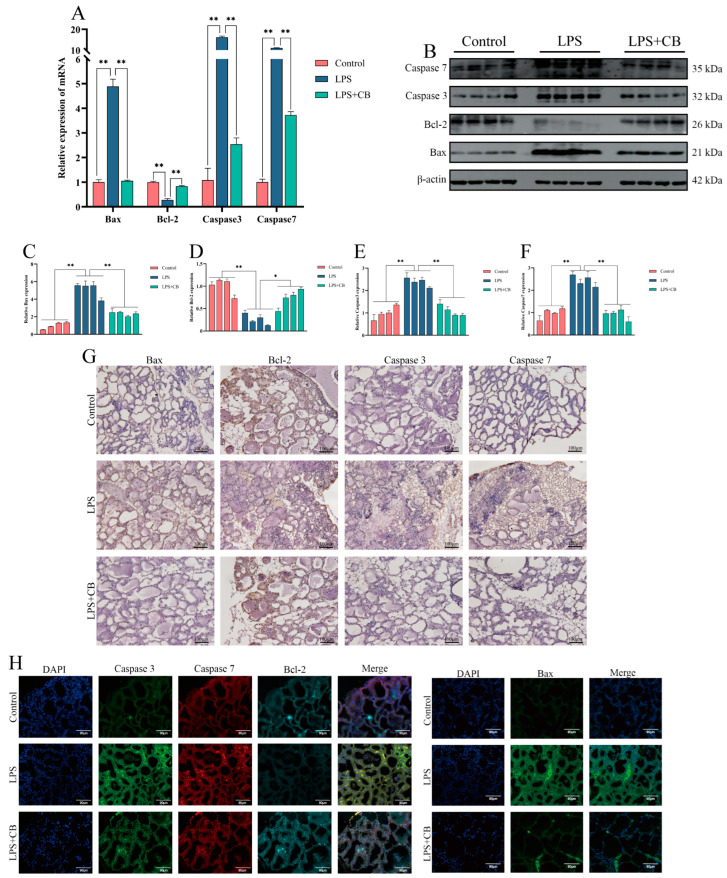
CB inhibits LPS-induced apoptosis of mouse mammary gland cells. (**A**) The mRNA levels of Caspase 3, Caspase 7, Bax, and Bcl-2 in mouse mammary gland were detected by qPCR. (**B**–**F**) Western blot was used to detect the expression of Caspase 3, Caspase 7, Bax, and Bcl-2 protein in breast tissue of mice. (**G**) IHC was used to detect the expression of Caspase 3, Caspase 7, Bax, and Bcl-2 protein in mouse breast tissue. Scale bar: 100 μm (black line segment). (**H**) IF was used to detect the expression of Caspase 3, Caspase 7, Bax, and Bcl-2 protein in mouse breast tissue. Scale bar: 90 μm (white line segment). *, *p* < 0.05; **, *p* < 0.01.

**Figure 9 ijms-26-03029-f009:**
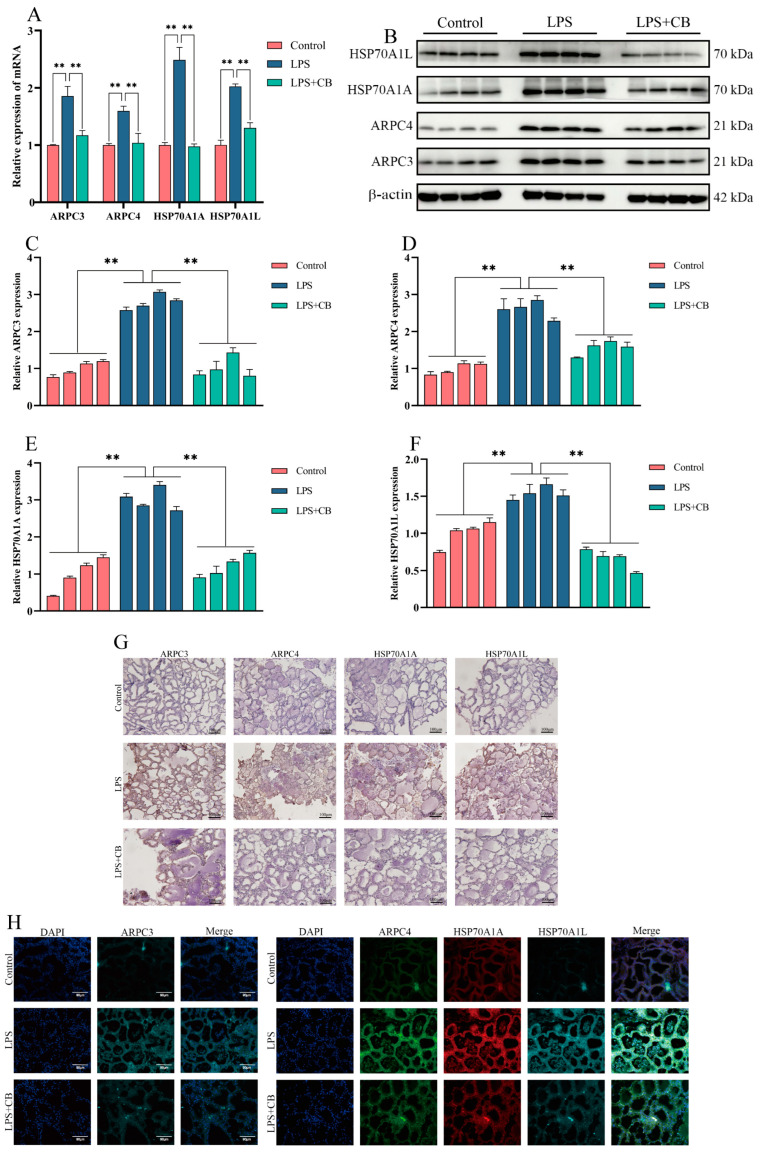
CB inhibited LPS-induced expression of ARPC3, ARPC4, and HSP70 in mouse mammary gland. (**A**) qPCR was used to detect the expression of ARPC3, ARPC4, HSP70A1A, and HSP70A1L mRNA in mouse breast tissue. (**B**–**F**) Western blot was used to detect the expression of ARPC3, ARPC4, HSP70A1A, and HSP70A1L protein in mouse breast tissue. (**G**) IHC was used to detect the expression of ARPC3, ARPC4, HSP70A1A, and HSP70A1L proteins. Scale bar: 100 μm (black line segment). (**H**) IF was used to detect the protein expression of ARPC3, ARPC4, HSP70A1A, and HSP70A1L. Scale bar: 90 μm (white line segment). **, *p* < 0.01.

**Table 1 ijms-26-03029-t001:** The information of all primers.

Species	Gene	GenBank NO.	Sequence (5′–3′)	Length
Mouse	IL-1β	NM_008361.4	GCCACCTTTTGACAGTGATGAG	135
			ATGTGCTGCTGCGAGATTTG	
	TNF-α	NM_013693.3	AAACCACCAAGTGGAGGAGC	120
			ACAAGGTACAACCCATCGGC	
	NLRP3	NM_145827.4	ATTACCCGCCCGAGAAAGG	83
			CATGAGTGTGGCTAGATCCAAG	
	BAX	NM_007527.4	CACTAAAGTGCCCGAGCTGA	96
			CAGCCACCCTGGTCTTGG	
	Bcl-2	NM_009741.5	GAACTGGGGGAGGATTGTGG	194
			GCATGCTGGGGCCATATAGT	
	Caspase 3	NM_009810.3	GGAGCAGCTTTGTGTGTGTG	242
			AGCCTCCACCGGTATCTTCT	
	Caspase 7	NM_007611.3	GAGGAGGACCACAGCAACTC	238
			CGTCAATGTCGTTGATGGGC	
	ARPC3	NM_019824.4	GCCATTTATGCCAAGCCTGC	151
			TCACAAAGCAAGTCCACCACT	
	ARPC4	NM_026552.3	CTGCCACTCTCCGCCCCTAC	126
			TGCTACTCCTGACTTCGACCTCTG	
	HSP70A1A	NM_005345.6	GACAAGTGCCAGGAGGTCAT	155
			CCGAAGCCCCCAGCC	
	HSP70A1L	NM_013558.2	GACGCCAACGGTATCCTGAA	176
			TTGGCAGCGATTTTCTCCCT	
	β-actin	NM_007393.5	GGCTGTATTCCCCTCCATCG	154
			CCAGTTGGTAACAATGCCATGT	
Bovine	IL-1β	NM_174093.1	TCCGACGAGTTTCTGTGTGA	206
			ATACCCAAGGCCACAGGAAT	
	TNF-α	NM_173966.3	TTGTTCCTCACCCACACCAT	239
			CCAAAGTAGACCTGCCCAGA	
	NLRP3	NM_001102219.1	GGCACCTTTCTTCCATGGCT	219
			ACCCGGTCAGAGTCCAGAAA	
	BAX	NM_173894.1	GAGATGAATTGGACAGTAACA	118
			TTGAAGTTGCCGTCAGAA	
	Bcl-2	NM_001166486.1	ATGACCGAGTACCTGAAC	79
			CATACAGCTCCACAAAGG	
	Caspase 3	NM_001077840.1	AGTGGTGCTGAGGATGAC	135
			ACAAAGAGCCTGGATGAA	
	Caspase 7	XM_002698509.6	GAAATTCAGCCTGCTTCGCC	110
			CCCCCTAAAATGGGCTGTCA	
	ARPC3	NM_001034271.2	TTTTCGTTGGGGTGGAGACO	138
			TCTCTAGGGGCAGGTCCTTI	
	ARPC4	NM_001076163.1	CCGTGTCTCTGTGAAGTCGT	103
			ATCTAATGCCCACCCTGACC	
	HSP70A1A	NM_203322.3	AGTGCCAGGAGGTGATTTCC	100
			ATGGGGTTACACACCTGCTC	
	HSP70A1L	NM_001167895.1	GCCAAGAACCAGGTAGCCAT	149
			ATTACCTTGGGCTTGCCTCC	
	β-actin	AY141970.1	CAACCGTGAGAAGATGACCCA	293
			TGTCACGGACGATTTCCGCTC	

**Table 2 ijms-26-03029-t002:** The information of all antibodies.

Name	Manufacturer	Cat.NO.
IL-1β	Affinity	AF4006
TNF-α	Affinity	AF7014
NLRP3	Proteintech	27458-1-AP
BAX	Proteintech	50599-2-IG
Bcl-2	Proteintech	26593-1-AP
Caspase 3	Bioss	bs-0081R
Caspase 7	Bioss	bsm-60304R
ARPC3	Proteintech	14652-1-AP
ARPC4	Proteintech	10930-1-AP
HSP70A1A	Proteintech	10995-1-AP
HSP70A1L	Proteintech	13970-1-AP
β-actin	Proteintech	66009-1-IG

## Data Availability

The datasets analyzed or generated during this study are available from the authors (d.wt2008@163.com).

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
