# Peer review of "Cytochalasin B Mitigates the Inflammatory Response in Lipopolysaccharide-Induced Mastitis by Suppressing Both the ARPC3/ARPC4-Dependent Cytoskeletal Changes and the Association Between HSP70 and the NLRP3 Inflammasome"

_ijms, 2025, doi:10.3390/ijms26073029_

Round 1

Reviewer 1 Report

Comments and Suggestions for Authors

First, I would like to appreciate the authors for their contribution in exploring the role of Cytochalasin B in mitigating LPS induced inflammatory response in Mastitis. The correlation between ARP2/3 complex, HSP70 and NLRP3 inflammasome was well studied and the authors have very well demonstrated the interrelationship by using various in vitro and in vivo models. However, I would like to pose some questions to the authors.

  1. One main aspect I would like to point out was that the authors haven't used an inflammatory cell system to demonstrate the effects of CB, If the authors could explain the reasoning behind using just the bovine mammary epithelial cells but not a macrophage cell line would be beneficial to the readers
  2. Have the authors looked into performing any migration assays as CB is actively involved with actin-related protein 2/3 complex
  3. Does CB effect the levels of N-WASP? It would be interesting to see how it influences N-WASP
  4. There was an uptrend observed in the levels of inflammatory markers observed in control groups across figures 1,2,5,6,8 and 9. Is this trend observed due to sample handling or was there an actual trend to be concerned about?
  5. Have the authors also looked into any other inflammatory pathways such as NF-kb/MAPK/JNK ? 
  6. The images were blurry and it was hard to read the figure legends, I would like to request the authors to provide high resolution images to help the readers.
  7. Where was CB procured from ? and also I would like to ask the authors about the script mentioned in different parts of the manuscript (line 111,134,146 etc)

Reviewer 2 Report

Comments and Suggestions for Authors

Some considerations must be cleared:

  1. Limited Scope of Models. The study primarily relies on in vitro (MAC-T cells) and in vivo (mouse) models. While these models are informative, results may not fully translate to clinical practice in dairy cows due to species differences and the complex nature of mastitis in a real-world farm setting.
  2. Lack of Longitudinal Data. The study may not provide sufficient longitudinal data on the long-term effects of CB treatment, which is crucial for assessing its safety and efficacy over time, especially in a livestock context.
  3. Dosing and Administration. Details regarding the optimal dosing regimen of CB and its pharmacokinetics in bovines are essential for practical application but may not be fully addressed in the study. Further research is necessary to determine the effective and safe dosage in dairy cows.
  4. Mechanistic Understanding. While the study suggests mechanisms involving the disruption of cytoskeletal rearrangement and inflammasome activation, more detailed mechanistic studies would be necessary to fully elucidate these pathways and to rule out any off-target effects of CB.
  5. Control and Comparisons. The article mentions using control groups, but further elaboration on the specific controls or comparisons against existing treatments (like antibiotics) would strengthen the argument for CB as a viable alternative.
  6. Economic Impact Analysis. While it highlights the economic burden of mastitis on dairy farming, a detailed cost-benefit analysis of introducing CB as a treatment compared to existing methods might provide further insights into its practical application.

Some Chinese letters in lines 111, 124, 134, 146, 159, 170, 187, 205, 219
